# Effective for Whom? A Review of Psychological and Social Intervention Recommendations in European Dementia Care Guidelines Through the Lenses of Social Health and Intersectionality

**DOI:** 10.3390/bs15040457

**Published:** 2025-04-02

**Authors:** David Neal, Sara Laureen Bartels, Saloua Berdai Chaouni, Thais Caprioli, Adelina Comas-Herrera, Rabih Chattat, Ana Diaz, Rose-Marie Dröes, Thomas Faulkner, Simone Anna Felding, Manuel Franco-Martin, Clarissa Giebel, Manuel Gonçalves-Pereira, Samira Hesse, Iva Holmerova, Wei Qi Koh, Emily Mena, Julia Misonow, Anahit Mkrtchyan, Nicole Müller, Martina Roes, Isabeau van Rompuy, Joanna Rymaszewska, Dorota Szcześniak, Jochen René Thyrian, Marjolein de Vugt, Amy Walden, Karin Wolf-Ostermann, Louise Hopper

**Affiliations:** 1Department of Medical Informatics, Amsterdam UMC, 1100 DD Amsterdam, The Netherlands; 2Alzheimer Centrum Limburg, Department of Psychiatry and Neuropsychology, Mental Health and Neurosciences Research Institute, Faculty of Health Medicine and Life Sciences, Maastricht University, 6229 ET Maastricht, The Netherlands; sara.bartels@maastrichtuniversity.nl (S.L.B.); isabeau.vanrompuy@maastrichtuniversity.nl (I.v.R.); m.devugt@maastrichtuniversity.nl (M.d.V.); 3Society and Ageing Research Lab, Department of Adult Educational Sciences, Vrije Universiteit Brussel, 1050 Brussels, Belgium; saloua.berdai-chaouni@vub.be; 4Department of Primary Care & Mental Health, University of Liverpool, Liverpool L69 3GF, UK; t.caprioli@liverpool.ac.uk (T.C.); t.faulkner@liverpool.ac.uk (T.F.); clarissa.giebel@liverpool.ac.uk (C.G.); 5NIHR Applied Research Collaboration North West Coast, Liverpool L69 3GL, UK; 6Care Policy and Evaluation Centre, London School of Economics and Political Science, London WC2A 2AE, UK; a.comas@lse.ac.uk; 7Department of Psychology, University of Bologna, 40126 Bologna, Italy; rabih.chattat@unibo.it; 8Alzheimer Europe, L-1736 Luxembourg, Luxembourg; ana.diaz@alzheimer-europe.org; 9Department of Psychiatry, Amsterdam University Medical Centers, 1081HJ Amsterdam, The Netherlands; rm.droes@amsterdamumc.nl; 10Mersey Care NHS Foundation Trust, Liverpool L34 1PJ, UK; 11German Center for Neurodegenerative Diseases (DZNE), 58453 Witten, Germany; simoneanna.felding@dzne.de (S.A.F.); anahit.blog@gmx.de (A.M.); martina.roes@dzne.de (M.R.); 12Department of Nursing Science, Faculty of Health, University of Witten/Herdecke, 58448 Witten, Germany; 13Department of Psychiatry and Mental Health, Hospital Provincial de Zamora, Salamanca University, 49071 Zamora, Spain; mfranco@usal.es; 14NOVA Medical School, Faculdade de Ciências Médicas, Universidade NOVA de Lisboa, 1169-056 Lisbon, Portugal; gpereira@nms.unl.pt; 15Comprehensive Health Research Center, Associated Laboratory REAL (LA-REAL), 1150-190 Lisbon, Portugal; 16Department of Developmental Psychology, Tilburg University, 5000 LE Tilburg, The Netherlands; h.s.hesse@tilburguniversity.edu; 17Centre of Expertise in Longevity and Long-Term Care, Charles University, 18200 Prague, Czech Republic; iva.holmerova@gerontocentrum.cz; 18School of Health and Rehabilitation Sciences, The University of Queensland, 4072 Brisbane, Australia; weiqi.koh@uq.edu.au; 19Institute for Public Health and Nursing Research & High-Profile Area of Health Sciences, University of Bremen, 28359 Bremen, Germany; e.mena@uni-bremen.de (E.M.); jmisonow@uni-bremen.de (J.M.); wolf-ostermann@uni-bremen.de (K.W.-O.); 20Department of Speech & Hearing Sciences, University College Cork, T12 K8AF Cork, Ireland; nicole.muller@ucc.ie; 21Department of Clinical Neurosciences, Faculty of Medicine, Wroclaw University of Science and Technology, 51-377 Wroclaw, Poland; joanna.rymaszewska@pwr.edu.pl; 22Department of Psychiatry, Medical Faculty, Wroclaw Medical University, 50-367 Wroclaw, Poland; dorota.szczesniak@umw.edu.pl; 23German Center for Neurodegenerative Diseases (DZNE), 17489 Greifswald, Germany; rene.thyrian@dzne.de; 24Institute for Community Medicine, University Medicine Greifswald, 17489 Greifswald, Germany; 25School of Psychology, Dublin City University, D09 N920 Dublin, Ireland; amykatewalden@gmail.com (A.W.); louise.hopper@dcu.ie (L.H.)

**Keywords:** intervention, dementia, social health, intersectionality, guideline, diversity, equity

## Abstract

In dementia care, access to effective psychosocial interventions is often addressed by evidence-based guidelines for care providers. However, it is unclear if current guidelines consider personal characteristics that may impact intervention effectiveness. This study investigates if, and within what framing, dementia care guidelines in Europe address what is effective and for whom. A review of 47 guidelines from 12 European countries was conducted. Content analysis focused on (i) if guidelines recommended specific psychosocial interventions, and how guidelines referred to (ii) social health, (iii) the intersection of social positioning, and (iv) inequities in care or outcomes. Thirty-five guidelines (74%) recommended specific psychosocial interventions. Around half referenced aspects of social health and of intersectionality. Thirteen guidelines (28%) referenced inequities. Social health was not explicitly recognised as a mechanism of psychosocial interventions. Only age and comorbidity were consistently considered to impact interventions’ effectiveness. Inequities were acknowledged to arise from within-country regional variations and individual economic status, but were not linked to (intersectional) individual societal positions such as sex and/or gender, sexuality, and/or religion. The results between European countries were heterogeneous. Current guidelines offer little insight into what works for whom. Policymakers and guideline developers should work with researchers, generating and translating evidence into policy.

## 1. Introduction

Dementia is a public health priority in Europe and worldwide ([2]; [53]). The 2024 update of the *Lancet Commission* provides hopeful evidence on dementia prevention ([27]). Moreover, biomedical research has resulted in diagnostic approaches that may be precise and timely ([32]; [44]). However, there is no cure for dementia. The newest pharmacological agents are not cost-effective, having only a modest effect on underlying disease progression and quality of life, despite, in some cases, causing serious side effects and high costs to health care systems ([23]; [31]). Meanwhile, there is evidence that social support can be a protective factor for mortality in people with dementia ([5]), and a number of specific non-pharmacological, psychosocial interventions can effectively improve, for instance, everyday functioning, depression and anxiety, and quality of life of people living with dementia without noticeable side effects ([33]; [28]; [41]). Improvements in social health—that is, the facilitation of self-management, social participation, and/or the ability to fulfil one’s obligations in society ([12]; [49])—may therefore be both an important end in and of itself, as well as an important mechanism mediating the effectiveness of psychological and social interventions on practical, mental, and social functioning.

Since psychological and social therapies in dementia may result in more meaningful benefits for the individual than currently available medications ([6]; [26]), they should be available and accessible for everyone. An important step for improving access is translating research into guidelines on managing dementia for health and care providers, such as primary care and specialist physicians, nurses, allied health professionals, and social care professionals ([47]; [25]; [50]). If guidelines provide clear recommendations on which therapies are effective, this can facilitate professionals in providing evidence-based referrals or (social) prescriptions ([39]). However, many guidelines on managing dementia are written within a biomedical paradigm by authors with primarily biomedical expertise, and in some cases, guidelines are funded by industry, introducing potential conflicts of interest ([51]; [43]; [14]). This raises the question of whether evidence on the effectiveness of psychological and social interventions, potentially mediated by improvements in social health, is appropriately identified, interpreted, valued, and translated into recommendations within professional guidelines for dementia management in different European countries. Our hypothesis is that this side of dementia management is frequently overlooked.

To ensure equitable care, treatment guidelines need to offer professionals recommendations that not only reflect what is effective but also for whom interventions are effective. As delineated in the Dementia Inequalities Model ([16]), inequitable care can arise due to inadequate policy and practices for addressing inequalities, for example, in dementia diagnosis, and care access and outcomes, at three layers: (i) ignoring personal characteristics, such as age, sex and gender, ethnicity, dementia subtype, and digital literacy; (ii) non-tailored social and community networks, such as lack of availability of a carer and peer support; and (iii) non-tailored society and health care infrastructure, such as lack of health and social care workforce knowledge and integration and lack of available and tailored services. Therefore, available interventions need to be considered in light of these factors. A lens of intersectionality can help to illuminate personal characteristics (i.e., sex, gender, and sexual preferences) and contextual factors (i.e., socio-economic, environmental, and societal) that may determine whether or not an intervention is effective for a particular person or group in society ([37]). Solving social problems within a given local, regional, national, or global context requires intersectional analyses. For example, an intersectional lens provides insights into how power relations (e.g., providing access or not) and discursive definitions (e.g., societal status) are represented in social hierarchies and lead to the privilege of any group to disadvantage another. An intersectional lens therefore provides insights into how different characteristics intersect, although intersectionality points out that not one intersection or social location is more important than another (equal perspective) ([24]). Analysing these (power) relations helps to understand how social positioning of an individual person converges in order to create unique forms of exclusion manifested in the development and application of the intervention ([9]). To the best of the authors’ knowledge, it is currently unclear whether recommendations for effective psychological and social therapies in national guidelines on dementia care address intersectionality or account for aspects of inequity as well as inequality.

The goal of this review was to identify if, and within what framing, existing national guidelines for the management of dementia in different European countries provide clear recommendations on which psychological and social therapies are effective, and for whom. Specifically, three conceptual lenses of social health, intersectionality, and inequity were applied to explore how these concepts manifest in European guidelines. Findings of this study will inform recommendations of the European Joint Programme on Neurodegenerative Diseases-funded INTEREST network for future research, practice, and policymaking to improve dementia care across Europe.

## 2. Materials and Methods

### 2.1. Study Design

Members of the Joint Programme on Neurodegenerative Diseases-funded INTEREST project conducted a cross-sectional narrative review of national guidelines on the management of dementia in Europe. The report is in line with the SANRA checklist for narrative reviews ([3]).

### 2.2. Identification of Guidelines

Between April and June 2024, guidelines were identified by purposely sampling documents from countries in which members of the INTEREST group were professionally active, and with which they were familiar, as experts in the field. Purposive sampling of guidelines was undertaken rather than a systematic search owing to resource limitations. Guidelines were eligible for inclusion if they provided specific recommendations to individual health and care professionals or to provider organisations on diagnosis, medical treatment, and care and support of people living with dementia and were supported by a non-profit or public organisation without apparent commercial interest linked to the recommendations of the guideline. This sampling approach resulted in at least one guideline being identified from each of the 12 countries involved in INTEREST, which are collectively home to approximately half of the population of Europe ([13]; [46]): Austria, Belgium, Czech Republic, Denmark, Germany, Ireland, Italy, the Netherlands, Poland, Portugal, Spain, and the United Kingdom. In total, 47 guidelines from these countries were included for analysis, and full texts were retrieved for all of the identified guidelines. The Appendix A described the details of the included guidelines.

### 2.3. Data Extraction

Content analysis was performed on the included guidelines. In each case, the analysis was performed, and data were extracted by a native speaker of the language in which the guideline was written, using a form created for this study by DN. The form was piloted and then discussed with all group members, which did not result in changes. Data extraction was checked by a second, senior researcher in the field of psychological, social, nursing, or medical sciences, who, with the exception of the Danish guidelines, as only one Danish-speaking researcher was involved, was also a native speaker of the language in which the guideline was written.

The data extraction addressed (1) whether the guideline recommended specific, effective psychological or social therapies, (2) passages of text referencing social health or components of social health, (3) passages of text referencing intersectionality or any individual contextual factors considered within a framework of intersectionality, and (4) passages of text that addressed inequity or inequality. An overview of concepts and definitions as well as categories for data extraction is provided in Table 1.

### 2.4. Data Analysis and Synthesis

Counts were reported for the number of guidelines identified per country, the number of guidelines recommending specific psychological or social interventions, and the number of guidelines with, respectively, at least one reference to aspects of social health, intersectionality, or inequity. The results were tabulated, alongside the psychological and social interventions recommended, and broken down by country. A narrative synthesis of the passages extracted from the guidelines with respect to each of the three main concepts (social health, intersectionality, and inequity) was undertaken. Illustrative quotations from guidelines were translated into English for reporting using DeepL translation software, and translations were checked by native speakers. Finally, a relational analysis was performed, resulting in a concept map, to illustrate interrelationships between (aspects of) social health, intersectionality, and inequity.

### 2.5. Public Involvement

The findings of the review were presented and discussed with members of two existing working groups coordinated by Alzheimer Europe (the European Working Group of People with Dementia, EWGPWD, and the European Dementia Carers Working Group, EDCWG). This work was conducted within the framework of public involvement and, therefore, ethical approval was not required ([1]). Members of both groups are from different European countries and have (or are providing care to people with) different types of dementia and at different stages of the disease. Members of these European groups were nominated at a national level to participate for a fixed term. The sessions took place in December 2024 in Brussels and were facilitated by two researchers involved in the INTEREST project and members of staff of Alzheimer Europe. The meeting with the EWGPWD included a total of 11 people with dementia and their supporters (a person of their choice who provides support for travel and at meetings, usually a family member or friend), and the meeting with the EDCWG included a total of 13 informal carers. All members received information in advance about the issues and questions to be discussed at the meeting. Their feedback and comments have been included in the Discussion and Recommendation Sections of this article.

## 3. Results

### 3.1. Recommended Psychological and Social Interventions

In total, 35 of the 47 (73%) reviewed guidelines recommended specific psychological or social interventions for people with dementia. Table 2 shows the tabulated results of vote-counting from a conceptual content analysis and the nature of the psychological and social interventions recommended by at least one guideline per country for which guidelines were included. In summary, at least one psychological or social intervention was recommended in 11 out of 12 countries, although collectively, guidelines from some countries recommended more interventions than others.

In more than seven countries, (cognitive) behavioural interventions or modifications, cognitive rehabilitation, training, and stimulation therapy, music therapy, physical activity and exercise, reminiscence, and sensory stimulation (aroma, massages, touch, and light) were suggested for people with dementia. In three to six countries, (creative) art therapy, assistive technology/technological aids, counselling/psychotherapeutic interventions, dance therapy, environmental assessments, modifications, interventions, family/interpersonal therapy, life story work/review, occupational therapy-based interventions, personal validation/compassion therapy, pet-/animal-assisted therapy, psychoeducation, and speech and language therapy-based approaches were recommended. In guideline(s) from two countries, carer interventions, conversational coaching/communication training, drama therapy, horticultural therapy/therapeutic gardens, mindfulness, physiotherapy, reality orientation, and sleep hygiene were highlighted. Finally, in guideline(s) from one country, care planning, compensatory strategies, doll therapy, Meeting Centre Support Programmes, nutritional care, and yoga were mentioned. Guideline(s) also mentioned that some of these interventions could take place in groups or could be combined with each other.

Overall, few references to inequity were identified across any of the guidelines (n = 13). Just over half of the guidelines were identified as containing references to any aspect of social health (n = 27) or intersectionality (n = 28).

### 3.2. “Social Health” in Guidelines on Psychological and Social Interventions for Dementia

Mentions of the exact term “social health” (or equivalent translation) were very rare in the guidelines, only appearing in guidelines from Ireland and the Netherlands. The guidelines mainly referred to social health implicitly, in the context of self-management and social participation. The self-management comments mostly concerned either the importance of functional capabilities with respect to activities of daily living or the importance of autonomy in daily life and in shared decision-making with respect to care and therapies.


*“The Dementia Model of Care outlines pathways of care that promote autonomy, timeliness, outcome-focused, person-centred and citizenship approaches for people living with dementia;”*
[Appendix A, Guideline 17 (Ireland: “Model of Care for Dementia in Ireland”), p. 17]

Regarding social participation, the importance of interactions within a supportive network and the broader context of social structures were frequently discussed. However, in some cases, this was presented only as background information on key aspects of care and quality of life. Only in a few guidelines were social interactions discussed as an outcome of a specific recommended intervention.


*“The principles of person-centered care underpin good practice in the approach to people with dementia and their families. These principles state […] the importance of relationships and interactions with other people in promoting the health and well-being of the person…”*
[Appendix A, Guideline 33 (Portugal: “Norma nº 053/2011 atualizada a 21/04/2023: Abordagem Terapêutica das Alterações Cognitivas”), p. 15]

### 3.3. “Intersectionality” in Guidelines on Psychological and Social Interventions for Dementia

Similarly to social health, explicit references to the term “intersectionality” were lacking in the guidelines. There were also no indications that a truly intersectional lens was applied by guideline writers. Two patterns emerged from references to individual characteristics and social positions. First, the most commonly mentioned characteristics were age and comorbidities (or physical and mental ability with respect to social position). With respect to age, guidelines highlighted specific needs of people with young-onset dementia but also highlighted dementia as a disease associated with older age, and by extension, co-occurring with physical disabilities and other comorbidities that might render specific therapies more or less appropriate. Guidelines highlighted physical and mental ability in a negative framing of disability owing to comorbidities, both in that disability can negatively impact access to or outcomes of dementia care and that dementia may result in less equitable access to care for comorbid diseases. Some guidelines used these examples to underscore how specific health needs vary between individuals and the importance of person-centred or personalised care.


*“A high proportion of people living with dementia (72%) will also have multiple mental and physical health comorbidities, the most common of which are arthritis, hearing problems, heart disease, or a physical disability.”*
[Appendix A, Guideline 45 (United Kingdom: “The dementia care pathway: full implementation guidance”), p. 7]


*“Services designed to meet the needs of younger people with dementia are likely to be more relevant and useful than similar services designed for older people.”*
[Appendix A, Guideline 46 (United Kingdom: “A guide to psychosocial interventions in early stages of dementia, second edition”), p. 73]

Second, many of the guidelines that did reference individual characteristics that may align with social positions also noted, at a higher level, that many of these are protected characteristics (key dimensions such as gender and ethnicity), on the basis of which there should be no unfair discrimination in providing care. For example,


*“There are groups of people with dementia who have very specific needs. These include younger people with dementia, people with sparse social networks, people from ethnic minorities, and people with dementia and developmental disabilities who differ from the general population of people with dementia in terms of diagnosis, treatment, care, and disease course.”*
[Appendix A, Guideline 10 (Denmark: “Anbefalinger for tværsektorielle forløb for mennesker med demens”), p. 11]

In some cases, there was also explicit discussion of stigma arising from a dementia diagnosis, with or without the intersectionality of these additional identities, but in most cases, there were no implications or recommendations with respect to specific therapies tied to these discussions.

### 3.4. “Inequity” in Guidelines on Psychological and Social Interventions for Dementia

The concept of inequity was primarily represented through references to within-country regional variations in service provision (Ireland, Netherlands, Portugal, and UK), in some cases with respect to specific therapies or services. This was sometimes further specified as relating to concerns about shortages of qualified staff.


*“The availability of activity programs tailored to people with dementia varies by region and healthcare institution.”*
[Appendix A, Guideline 27 (the Netherlands: “Probleemgedrag bij mensen met dementie”), p. 118]

In guidelines from Italy, Portugal, and the UK, direct references were found to other sources of inequity, with statements emphasising the need for access to care regardless of, for example, ethnic background, socio-economic status, or comorbidity. However, guidelines mentioned these inequities mostly at a high level, rather than with respect to any specific therapy.


*“Health professionals should be aware of the need to ensure equitable access to treatment for patients from different ethnic groups and people from different cultural backgrounds.”*
[Appendix A, Guideline 23 (Italy: “Diagnosi e trattamento di demenza e Mild Cognitive Impairment”), p. 278]

Guidelines largely omitted any acknowledgment or discussion of the (un)representativeness of primary research samples on which recommendations were based.

### 3.5. Relational Analysis: Concept Mapping

Relational analysis exploring relationships between (components of) the concepts of social health, intersectionality, and inequity, as they are represented in the guidelines, resulted in a concept map (Figure 1). In summary, the social participation and self-management domains of social health were frequently implicitly referenced, as guideline authors acknowledged the importance of social networks, autonomy, and shared decision-making. However, social health was only rarely explicitly referenced, and not in the context of a mechanism of action of psychological and social interventions. Similarly, the concept of intersectionality was not explicitly referenced. Whilst a truly intersectional approach was not evident in guidelines, age and ability (referenced mostly with respect to comorbidities) were commonly acknowledged in isolation as important individual characteristics that might influence appropriateness of particular psychological or social interventions. Other personal characteristics were only acknowledged at a high level, existing as “protected characteristics”. There was no actual intersection acknowledged between these characteristics and inequities in any of the included guidelines, despite many guidelines implicitly acknowledging a theoretical potential for inequities arising from discrimination on the basis of such characteristics, through a normative expression that such characteristics should not be sources of inequity. Within-country regional variations in the availability of interventions and individual economic status in relation to out-of-pocket payments for care were acknowledged as actual sources of inequity with respect to accessing specific psychological and social interventions.

## 4. Discussion

This review aimed to identify whether, and with what framing, existing national guidelines for the management of dementia in 12 European countries provide clear recommendations on which psychological and social therapies are effective, and for whom. Specifically, the lenses of social health, intersectionality, and inequity were applied to explore the level of detail and specificity applied in European guidelines. Firstly, our findings reveal that of 47 reviewed guidelines, 74% (35 guidelines) recommended specific psychological and social interventions. A wide range of psychological and social interventions were recommended, with (cognitive) behavioural interventions, cognitive rehabilitation, training, and stimulation therapy, music therapy, physical activity and exercise, reminiscence, and sensory stimulation (aroma, massages, touch, and light) being most common. Secondly, only around half of the reviewed guidelines (60%, n = 28) contained references to aspects of social health. The importance of social health as a determinant of quality of life is often recognised in the guidelines, but this recognition is often implicit and high-level. The importance of the social network and contacts, as well as autonomy and shared decision-making, was highlighted, linking to two of the three social health domains, namely social participation and self-management, respectively. However, the social health domain, “the ability to fulfil one’s obligations in society”, was notably absent from the guidelines. Additionally, social health as a mechanism of impact of specific psychological and social interventions was rarely discussed. Thirdly, aspects of intersectionality were referenced in around half of the included guidelines (57%, n = 27), with age and comorbidities reported as potentially influencing the appropriateness of specific psychological and social interventions. Finally, very few guidelines (28%, n = 13) referenced inequities. Specifically, it was highlighted that within-country regional variations in care provision exist and that economic status may result in barriers to accessing some interventions. High-level discussions also mentioned that inequity “should not” arise from gender, ethnicity, religion, and cultural background, but specific recommendations to avoid inequity due to these personal characteristics were lacking such as, for example, recommendations detailing how these differences might impact the effectiveness of interventions or consideration of the representativeness of research samples from studies of intervention effectiveness (what is effective for whom). The present findings are discussed below, including recommendations for future research, policy, and practice.

### 4.1. Towards a Biopsychosocial Approach to Dementia Care Across Europe

Historically, dementia guidelines focused primarily on biomedical aspects such as blood tests, imaging, and cognitive assessments with little attention paid to psychological and social interventions ([52]; [51]; [30]). Thus, it is essential for the call to action to reduce the divide between biomedical and psychosocial dementia research, policy, and practice ([48]) to be reflected to some extent in a majority of reviewed European guidelines. However, 27% of the guidelines included in the present study still did not mention specific psychological and social interventions. Moreover, the level of detail between guidelines varied greatly. These findings indicate the need for further improvement across Europe. All of the guidelines that we reviewed from Denmark (4/4), Italy (1/1) Portugal (3/3), and the UK (5/5) included recommendations regarding specific psychological and social interventions, and the approach in these countries might serve as an example for refining guidelines that currently do not refer to these interventions. In addition, it would be important for research to capture the extent to which these recommendations have been implemented in practice. To improve dementia care in the future, all countries in Europe should be encouraged to develop evidence-based guidelines for health and care providers where these do not exist. In doing so, and in revising guidelines in countries where such guidance already exists, guideline developers should be sure to include recommendations to provide effective psychological and social interventions as part of a gold standard for multidisciplinary dementia care. Implementation of guidelines in practice should be monitored, and providers should be supported with information on facilitators and barriers for achieving successful guideline implementation.

### 4.2. Fragmented Use of Conceptual Frameworks Regarding What Is Effective for Whom

Aspects of all concepts of interest, namely social health, inequity, and intersectionality, were referenced to some degree in the included guidelines, which highlights a certain level of detail and nuance in existing recommendations, with respect to which psychological and social interventions are effective for whom. However, a large gap remains between the comprehensiveness of these conceptual frameworks and their piecemeal translation into guidelines.

For example, the omission of the social health domain “capacity to fulfil one’s potential and obligations” ([12]) highlights a critical gap in acknowledging how dementia may impact an individual’s capacity to function in society according to their competencies and talents, while also meeting social demands and how psychosocial interventions may support this. However, research on interventions that address this domain of social health is limited, and more evidence exists on interventions improving self-management and social participation ([28]; [36]). One review on dignity, which is linked to the fulfilment of social roles, concluded that research on evidence of interventions was either inconclusive, lacked rigour, or identified no long-term effects ([45]). Thus, the outcome of the review appears to mirror the limited evidence in the field. Moreover, while the other two social health domains (self-management and social participation) were mentioned, there was no explicit acknowledgement of the role of these aspects of social health as mechanisms of impact of specific interventions. Frameworks for elucidating mechanisms of impact should be utilised explicitly by researchers when developing, evaluating, or implementing complex interventions ([42]), but also by policymakers when creating guidelines for practice. This way, guideline recommendations can be grounded in theory as well as evidence. When guideline developers review and revise their guidelines, they can efficiently evaluate and include evidence for the effectiveness of new interventions, in light of the effects of those interventions on social health domains, as established mechanisms of impact for improving quality of life. Indeed, one of the utilities of this review is to help and contribute to defining the roadmap for updating dementia care guidelines.

Similarly, while certain personal characteristics or social positions such as age, ability, and comorbidity were mentioned in connection with the appropriateness of specific psychological and social interventions, the intersectionality lens as a whole was not applied to recommendations by guideline developers. This gap may partly reflect broader trends in the research context over the years, where diversity aspects have increasingly gained attention (often driven by funding body requirements), but the intersections between these aspects are still rarely addressed. The relatively recent emerging focus in research on how diversity characteristics intersect and interact with social systems and institutions in the context of dementia care likely contributes to the absence of a robust intersectionality perspective in current dementia care guidelines. However, the guidelines also demonstrate little acknowledgement of the well-established differences between the needs of individuals experiencing young- and late-onset dementia ([29]; [20]), as well as different dementia subtypes, such as behavioural frontotemporal dementia ([11]), posterior cortical atrophy ([19]), or Lewy body dementia ([4]), and this should also be addressed. To be able to deliver personalised and person-centred care, it is essential to understand not only what works, but what works for whom. Guideline developers should prioritise integrating findings from future intersectionality-focused research to make recommendations that inform health and care providers how to provide effective, person-centred psychological and social interventions.

### 4.3. Inequity in Dementia Care Arising from Differences in Guideline and Service Provision

This review demonstrates that inequity with respect to non-tailored access and outcomes of dementia care is insufficiently addressed in current guidelines. Whilst within-country regional differences and financial accessibility were acknowledged in some guidelines as sources of inequity, no recommendations were identified for how to address these concerns. Similarly, whilst age and ability were sometimes mentioned as impacting the access to or appropriateness of interventions, guidelines failed to acknowledge inequity arising from exclusionary mechanisms based on social positions (such as sex and/or gender, ethnicity or language, and socio-economic factors). However, we did not analyse the development process of the included guidelines; thus, based on our analysis, it was not our aim to analyse existing power relations within the participating disciplines that developed the guidelines. Therefore, conclusions about structural exclusionary mechanisms in the process of guideline development can only be inferred from our results. Based on the high relevance of guidelines in our health care systems, it is also important to raise awareness of sources of inequity and inequality in dementia care research, due to the fact that these results will be integrated in guideline development and implementation, which is also a potential source of inequity that was beyond the scope of this research to investigate ([18]).

These findings are concerning and are in line with another recent review (not connected to psychological and social interventions) that found 13 dementia guidelines with specific references to inequity, also highlighting heterogeneity ([17]). The authors’ conclusion that few dementia guidelines included tangible objectives towards reducing inequities is in line with the present study.

Moreover, these reviews demonstrate that, at the European level, inequity may arise owing to unwarranted heterogeneity between countries in guideline recommendations. If effective psychological and social interventions are recommended in some countries but not in others, this may exclude subgroups of people with similar needs from accessing effective care. Examples of this potentially exclusionary heterogeneity are, for instance, that carer interventions were only mentioned in few guidelines, even though evidence exists that supporting the well-being of families, partners, and children of the person with dementia is an essential pillar of dementia care ([8]; [15]; [38]). Similarly, assistive technologies are only included in guidelines from Ireland, Spain, and the UK, indicating not only a lack of transfer from scientific evidence ([7]; [21]; [35]) into clinical guidelines, but also a contribution to inequity across Europe.

### 4.4. Public Involvement Perspectives

These findings were discussed with the people with dementia and family carers in a public involvement event. The main reflections of the members of the EWGPWD and EDCWG are summarised below:Psychosocial interventions are a topic of great relevance for people with dementia and carers. Although guidelines and recommendations are usually targeted to healthcare professionals and policymakers, it would be important for people affected by dementia to receive information about them to understand what is recommended and what they may be entitled to.In addition to structured dementia-specific psychosocial interventions, many people with dementia identify a need to participate in social activities or hobbies that are meaningful to them, on “their own terms” (i.e., as opposed to structured, organised, and planned activities which are often recommended in the guidelines). This could include, for example, attending a football match or a concert with a friend. This reflects the need to promote autonomy, choice, and existing social networks.Some European countries do not have any guidelines, and therefore, efforts should be made to transfer knowledge and best practices from countries that have developed guidelines to those where they do not exist or are poorly developed.People with dementia and carers recognised the importance of evidence-based guidelines, but it was also highlighted that in many countries, good guidelines exist but are not well implemented, owing to inadequate dissemination or funding. This can lead to inequity, for example, in cases where a specific intervention is recommended, but is available only in certain areas or to certain groups of people. More research into broader issues surrounding guideline implementation is needed.The topic of intersectionality is key and should be better addressed in the guidelines, as social positions such as age, sex and/or gender, type and stage of dementia, and other cultural aspects of care are very important to consider in the interventions.It is important that people with lived experience of dementia are involved in developing guidelines so that they reflect their priorities, needs, and values.

### 4.5. Recommendations for Future Research and Policymaking

In Table 3, a summary of recommendations mentioned above is provided to advance future research and policymaking. The recommendations target policymakers and committees commissioned to write guidelines for dementia care, as well as researchers and health and care providers.

### 4.6. Strengths and Limitations

This review included a large number of dementia care guidelines, written in different languages, for health and care providers in countries with diverse demographics from across the European region. The results were interpreted with input from an international, multidisciplinary team of experts in the field of dementia care. Moreover, whilst not a representative sample of all people living with dementia, in itself a reflection of the impact of intersecting exclusionary mechanisms, important input from experts of the experience of those who live with dementia and support people living with dementia helped to arrive at recommendations for future research, policy, and practice that are relevant to these stakeholders.

Intersectionality is inherently complex and multi-faceted in addressing interactions and power dynamics between individuals and social systems to understand how inequity arises from accumulative exclusionary mechanisms. The design of this review allowed for scoping the state of current guidelines with respect to aspects of a model of intersectionality. Future studies should build on this with richer and more detailed critical readings of dementia guidelines from an intersectional perspective. Another important limitation of the review design was the non-systematic approach to identifying relevant European guidelines. A convenience sample was taken from all 12 countries in which experts from the research team were active, using guidelines with which they were familiar (or to which they had contributed, in some cases), at a single point in time. This allowed the research to be carried out efficiently within the constraints of the available resources. However, the sampling procedure is expected to have introduced bias into the study. The likely direction of this bias is towards over-sampling guidelines in which the key concepts were present and well developed, since these concepts relate to the expertise and interests of those contributing to this review. This bias was highlighted by the insight from the public involvement process, where some countries that we did not include did not have any guidelines whatsoever. Given that the key concepts evaluated by this review were so underdeveloped in spite of any selection bias, our conclusions may be optimistic, and the urgency of our recommendations may be even greater than we have accounted for. We also did not collect data on the professions or qualifications of the guideline developers, although this might be expected to account for some of the observed variation in the results between countries and guidelines, and this should be investigated further. This cross-sectional study could be improved upon by prospectively and longitudinally reviewing the status of dementia care guidelines in all European countries. Such a guideline observatory would enable monitoring of potential inequities between European countries, the overall quality of evidence-based recommendations for health and care providers, and targeted support for improvements where required. A further methodological consideration is the inherently subjective nature of narrative synthesis and concept mapping, although the involvement of two researchers per guideline and diverse stakeholder perspectives in interpreting results is expected to have increased the rigour and representativeness of the recommendations and conclusions.

## 5. Conclusions

Whilst the majority of European dementia guidelines seem to recommend psychological or social interventions, grounding in the theoretical model of social health as a mechanism of impact is lacking. Few guidelines consider what works for whom and when, or the extent to which inequity in care and outcomes may arise from intersectional social positions. Within European countries, research and guideline development should be more targeted toward understanding how individual, community, and societal and infrastructure characteristics can impact the access to and effectiveness of psychological and social interventions, by either acting as barriers or facilitators, and how this may lead to health inequities. Guideline recommendations should be revised accordingly. Guideline developers should account for social health as a mechanism of impact of psychological and social interventions, because recognition of mechanisms is expected to help ensure efficient translation of future research into practice, with respect to interventions harnessing established, recognised mechanisms of impact. Given the poor cost-effectiveness of novel pharmacological agents and the growing prevalence of dementia in Europe, addressing heterogeneity in European dementia care guidelines in the extent to which they recommend effective psychological and social interventions should be a priority for European researchers and policymakers to ensure equitable access to effective care that supports good quality of life by optimising mental and social health.

## Figures and Tables

**Figure 1 behavsci-15-00457-f001:**
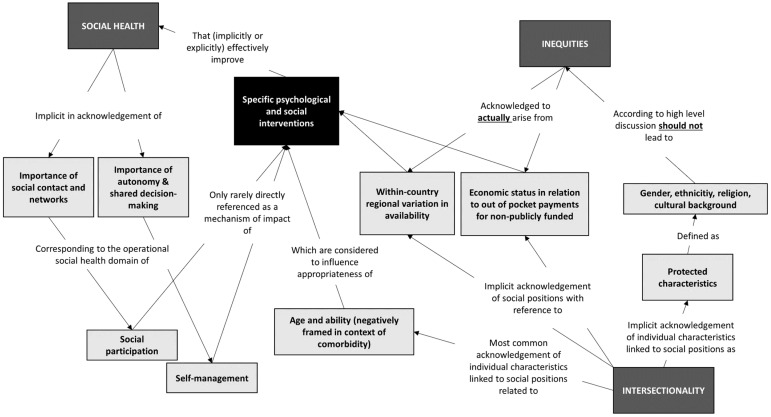
A concept map illustrating the relationship between (components of) the concepts of social health, intersectionality, and inequity, as they are represented in the guidelines analysed in this study.

**Table 1 behavsci-15-00457-t001:** Focus for data extraction including definitions of concepts and sub-categories.

Category	Definition and Sub-Categories for Data Extraction
Guideline characteristics	Country, title
Psychological and social interventions	Following [41] ([41]): Non-pharmacological interventions cover a diverse and broad range of intervention categories including, for instance, cognitive training, physical exercise, dietary treatments, art-oriented therapy, and reminiscence therapy ([40]); “any theoretically based, nonchemical, focused and replicable intervention, conducted with the patient or the caregiver, which potentially provided some relevant benefit” ([34]) Which generic recommendations about psychological or social interventions are made in the guideline? Which specific psychological and social interventions are recommended in the guideline?
Social health	Health is the ability to adapt and self-manage ([22]). Social health is specifically characterised by the following: (i) The capacity to fulfil one’s potential and obligations: The ability of a person (living with or caring for a person with dementia) to function in the society according to their competencies and talents (“potentials”) in the best possible way and to meet social demands (“obligations”) on a micro and macro societal level. (ii) Managing life with some degree of independence: The ability to manage life with some degree of independence can be operationalized as the ability to preserve autonomy and to solve problems in daily life, as well as to adapt to and cope with the practical and emotional consequences of dementia. (iii) Participation in social activities: The act of being occupied or involved with meaningful activities and social interactions and having social ties and relationships, which are meaningful to the person living with dementia themselves ([12]; [49]). How is social health explicitly referenced in the guideline? How are the three social health domains referenced?
Intersectionality	Intersectionality, as a theory and methodology, acknowledges the complexity and multidimensionality of people’s lives, and highlights that a person—due to their social positioning (e.g., socio-economic factors, sex and gender, and ethnicity)—may experience health-related stigma and other disadvantages ([9]; [37]; [24]). For this study, we focused on the intrapersonal or micro-level identities that can shape stigma and impact equitable access to high-quality care and health outcomes, as defined in [37] ([37]). How is intersectionality explicitly referenced in the guideline? How are the specific dimensions of intersectionality referenced?
Inequity/inequality	Inequities in a health care context are apparently avoidable or unjust differences in health status, the distribution of health determinants, or access to health and social care between different population groups ([10]). Inequitable care is the result of ignoring differences or inequalities in health status, the distribution of health services, or access to health and social care between different population groups ([9]; [37]). How is inequity explicitly referenced in the guideline? How is the (un)representativeness of samples in primary research referenced in the guideline?

**Table 2 behavsci-15-00457-t002:** Quantitative summary of guidelines identified by country, including an overview of which psychological and social therapies are recommended and how often social health, intersectionality, and inequity were referenced.

	AT	BE	CZ	DK	DE	ES	IE	IT	NL	PL	PT	UK	Total
**Guideline(s) identified (n)**	1	1	6	4	4	7	6	1	4	5	3	5	47
**Guideline(s) recommending psychological or social interventions (n)**	0	1	3	4	3	6	4	1	3	2	3	5	35
(Creative) art therapy							x			x		x	3
Assistive technology/aids/telecare						x	x					x	3
Care planning			x										1
Carer interventions (incl. behavioural)											x	x	2
(Cognitive) behavioural therapy-based intervention/modification					x	x	x	x	x		x	x	7
Cognitive rehabilitation (therapy) (also in groups)			x	x		x	x	x		x	x	x	8
Cognitive restructuring									x				1
Cognitive stimulation (therapy) (incl. Cogs club)			x	x	x	x	x	x	x		x	x	9
Cognitive training			x	x	x	x	x	x		x	x		8
Compensatory strategies						x							1
Conversational coaching/communication training							x				x		2
Counselling/psychotherapeutic interventions			x				x	x			x		4
Dramatherapy							x					x	2
Dance therapy				x	x		x					x	4
Doll therapy							x						1
Environmental assessment, modification, and interventions						x	x	x					3
Family/interpersonal therapy						x	x					x	3
Horticulture therapy/therapeutic gardens							x	x					2
Life story work/review					x		x					x	3
Meeting Centre Support Programme										x			1
Mindfulness					x		x						2
Music therapy				x	x	x	x	x	x	x		x	8
Nutritional care			x										1
Occupational therapy-based interventions			x					x	x	x	x		5
Personal validation/compassion therapy			x		x	x						x	4
Pet-/animal-assisted therapy					x		x			x		x	4
Physical activity, exercise, fitness, and psychomotor therapy (incl. supervised)			x	x	x	x			x	x		x	7
Physiotherapy			x							x			2
Psychoeducation (also for carers)		x	x	x		x			x		x		6
Reality orientation						x				x			2
Reminiscence therapy (incl. group format)			x	x	x	x	x	x	x	x	x	x	10
Sensory stimulation therapy (incl. aroma, touch, massage, light, bathing, and snoezelen)			x	x	x	x	x		x		x	x	8
Sleep hygiene				x								x	2
Speech and language therapy (incl. speaking, chewing, swallowing, and breathing exercises)							x			x		x	3
Yoga							x						1
**Different interventions recommended/country (n)**	N/A	1	13	10	12	15	22	10	9	12	11	18	
**Guideline(s) with reference(s) to social health (n)**	N/A	0	1	2	2	7	6	0	2	1	1	5	27
**Guideline(s) with reference(s) to intersectionality (n)**	N/A	0	1	2	3	7	6	1	1	1	2	4	28
**Guideline(s) with reference(s) to inequity (n)**	N/A	0	0	0	0	1	4	1	2	0	1	4	13

Notes: AT: Austria; BE: Belgium; CZ: Czech Republic; DK: Denmark; DE: Germany; IE: Ireland; IT: Italy; ES: Spain; NL: the Netherlands; PL: Poland; PT: Portugal; UK: the United Kingdom. Combinations of therapies were also recommended for individuals or groups.

**Table 3 behavsci-15-00457-t003:** Summary of recommendations resulting from the present guideline review.

Recommendations for future research and policymaking to improve equitable access to effective dementia care in Europe: All countries in Europe should be encouraged to develop evidence-based guidelines for health and care providers where these do not exist. Guideline writers should place recommendations regarding psychological and social interventions within the context of the appropriate theoretical model of social health as a mechanism of impact, as well as an end in and of itself. This is an essential step toward person-centred, holistic care provision in the context of evolving care and treatment options, including novel disease-modifying pharmacological therapies. These therapies should also be evaluated based on their impact on social health. Guideline writers should move towards applying an intersectional lens within recommendations, ensuring that effective psychological and social interventions are recommended, with clear indications of for whom they are effective and accessible, accounting for the impact of intersecting social positions on care needs, access, and outcomes. This is an essential step to transfer knowledge to health and care providers about how to provide effective, person-centred care and actively reduce the risk of inequities arising from a one-size-fits-all approach. Policymakers and guideline writers should be open about all potential sources of inequity in care and outcomes and develop specific strategies to address sources of inequity. Attention should be paid to issues related to the implementation and access to the interventions recommended by the guidelines, with adequate funding and support for care providers. People with lived experience should be involved in the development of the guidelines and their implementation. Information in lay terms should be available about the guidelines for people affected by dementia and their caregivers. Future studies are needed to investigate how psychological and social interventions can support the social health domain “capacity to fulfil one’s potential and obligations” when living with dementia. Longitudinal, systematic research is needed to monitor the translation of evidence into guidelines, and from guidelines into practice, as an important influence on quality of care. Future research should build on the present analysis to examine the extent to which current guidelines are applicable to people with mild cognitive impairment, as well as dementia, and the impact of language on access to care.

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
