# Peer review of "Effective for Whom? A Review of Psychological and Social Intervention Recommendations in European Dementia Care Guidelines Through the Lenses of Social Health and Intersectionality"

_behavsci, 2025, doi:10.3390/bs15040457_

Round 1

Reviewer 1 Report

Comments and Suggestions for Authors

This excellent and comprehensive review of dementia care guidelines will be an invaluable resource and hopefully will inspire more research and more effective implementation/translation into practice.

Just some minor comments-

  1. Do any of the guidelines specifically differentiate MCI from dementia and if so do the suggestions vary for these? If not, it might be good to suggest they do.
  2. How do the guidelines address issues with language- specifically those who do not speak the language of that country. 
  3. Are the guidelines "ready" for the disease modifying therapies and if not how should they address this- these drugs are here!

Author Response

This excellent and comprehensive review of dementia care guidelines will be an invaluable resource and hopefully will inspire more research and more effective implementation/translation into practice.

Just some minor comments-

  1. Do any of the guidelines specifically differentiate MCI from dementia and if so do the suggestions vary for these? If not, it might be good to suggest they do.

Response to reviewer: Thank you for this interesting question. We did not extract information from the guidelines concerning the stated relevance to people with MCI, in addition to people living with dementia. We have added the following passage to the discussion on page 20:

“Future research should build on the present analysis to examine the extent to which current guidelines are applicable to people with mild cognitive impairment, as well as dementia, and the impact of language on access to care.”

  1. How do the guidelines address issues with language- specifically those who do not speak the language of that country. 

Response to reviewer: Thank you for this question. We extracted as many references as we could find in the guidelines that explicitly addressed inequity and added to this any references to social identities as defined in Harleen Rai’s model of intersectionality in the context of dementia care, as reported in the methods. Language was not mentioned in relation to inequity. Language and ethnicity, which is one of the factors in Rai’s model, are closely related, and, as we report, ethnicity was also largely overlooked. We consider this to be a very interesting question, however, and have added the following to the discussion:

Page 18: “Similarly, whilst age and ability were sometimes mentioned as impacting access to or appropriateness of interventions, guidelines failed to acknowledge inequity arising from exclusionary mechanisms based on social positions (such as sex and/or gender, ethnicity or language, and socio-economic factors).”

Page 20: “Future research should build on the present analysis to examine the extent to which current guidelines are applicable to people with mild cognitive impairment, as well as dementia, and the impact of language on access to care.”

  1. Are the guidelines "ready" for the disease modifying therapies and if not how should they address this- these drugs are here!

Response to reviewer: Thank you for this interesting comment. In this review, the focus lay on psychological and social interventions rather than pharmacological therapies. Psychological and social interventions will remain complementary to disease modifying therapies. Indeed, for some individuals psychological and social interventions will remain the only option, as disease modifying therapies are currently not approved or available for many people living with dementia in Europe. The extent to which dementia care guidelines are ready for these treatments is an interesting but separate research question which goes beyond the scope of the current review. We have nonetheless added an acknowledgement of the importance of these developments to our recommendations on page 20:

“Guideline writers should place recommendations regarding psychological and social interventions within the context of the appropriate theoretical model of social health as a mechanism of impact, as well as an end in itself. This is an essential step toward person-centred, holistic care provision in the context of evolving care and treatment options, including novel disease-modifying pharmacological therapies. These therapies should also be evaluated based on their impact on social health.”

Reviewer 2 Report

Comments and Suggestions for Authors

Greetings.
First of all, thanks for your work.
Your work it's well builded. You followed the main structure of this kind of paper. The introduction was wide enough to reflect the state of the art. The methods are well descriped and the discussion it's well oriented.
I've some small suggestions to improve your work:
In study design: "cross-sectional review", but which type? Narrative? Systematic? Rapid? Which review guideline had you followed?
In data extraction: the text between line 187 and 200 it's not needed because you duplicate the information in the table below. Or you use this text or use the table not both.
In limitations: I think that you have a major limitation that you do not critical appraise. You write this in methods: "guidelines were identified by purposely sampling documents from countries in which members of the INTEREST group were professionally active, and with which they were familiar, as experts in the field". Why did you made this choice? It's not written anywhere the reason of this choice. I understand why but this is a major limitation and, related with the funding, even more, because the bias risk it's high. What have you done to limit this bias? It's not expliced. 
Once more, thanks for you work

Author Response

Greetings. First of all, thanks for your work.Your work it's well builded. You followed the main structure of this kind of paper. The introduction was wide enough to reflect the state of the art. The methods are well descriped and the discussion it's well oriented. I've some small suggestions to improve your work:
In study design: "cross-sectional review", but which type? Narrative? Systematic? Rapid? Which review guideline had you followed?

Response to reviewer: thank you for this question. We consider the review to be best described as a narrative review, with the caveat that the exclusive focus was on grey literature (guidelines) rather than traditional academic literature. Our report is in line with the SANRA checklist for narrative reviews (Baethge C, Goldbeck-Wood S, Mertens S: SANRA—a scale for the quality assessment of narrative review articles. Research Integrity and Peer Review (2019) 4:5

https://doi.org/10.1186/s41073-019-0064-8). We have added a clarification to the methods (page 5) concerning the study design:

“Members of the Joint Program on Neurodegenerative Diseases-funded INTEREST project conducted a cross-sectional narrative review of national guidelines on the management of dementia in Europe. The report is in line with the SANRA checklist for narrative reviews (Baethge et al., 2019).”

In data extraction: the text between line 187 and 200 it's not needed because you duplicate the information in the table below. Or you use this text or use the table not both.

Response to reviewer: thank you for this observation, we are happy to take the opportunity to reduce the length of the manuscript and improve readability. We have accordingly deleted this passage of text, referring readers to Table 1.

In limitations: I think that you have a major limitation that you do not critical appraise. You write this in methods: "guidelines were identified by purposely sampling documents from countries in which members of the INTEREST group were professionally active, and with which they were familiar, as experts in the field". Why did you made this choice? It's not written anywhere the reason of this choice. I understand why but this is a major limitation and, related with the funding, even more, because the bias risk it's high. What have you done to limit this bias? It's not expliced. Once more, thanks for you work.

Response to reviewer: Thank you for highlighting this important limitation of the study. We added to the methods to highlight that this choice was related to resource limitations:

Page 5: “Purposive sampling of guidelines was undertaken rather than a systematic search owing to resource limitations.”

To our discussion of this limitation on page 21, we also made additions to make more explicit the resulting bias, measures taken to minimize bias, and impact on interpretation of the results:

“A convenience sample was taken from all 12 countries in which experts from the research team were active, of guidelines with which they were familiar (or to which they had contributed, in some cases), at a single point in time. This, allowed the research to be carried out efficiently within the constraints of the available resources. However, the sample procedure is expected to have introduced bias into the study. The likely direction of this bias is towards over-sampling guidelines in which the key concepts were present and well-developed, since these concepts relate to the expertise and interests of the those contributing to this review. This bias was highlighted by the insight from the public involvement process that some countries that we did not include, do not have any guidelines whatsoever. Given that the key concepts evaluated by this review were so under-developed in spite of any selection bias, our conclusions may be optimistic and the urgency of our recommendations may be even greater than we have accounted for.”